# Artificial Intelligence and the Limitations of Information

**Paul Walton**

Capgemini UK, Forge End, Woking, Surrey GU21 6DB, UK; paulnicholaswalton@gmail.com;
Tel.: +44-13-0688-3140

**Abstract:** Artificial intelligence (AI) and machine learning promise to make major changes to the relationship of people and organizations with technology and information. However, as with any form of information processing, they are subject to the limitations of information linked to the way in which information evolves in information ecosystems. These limitations are caused by the combinatorial challenges associated with information processing, and by the tradeoffs driven by selection pressures. Analysis of the limitations explains some current difficulties with AI and machine learning and identifies the principles required to resolve the limitations when implementing AI and machine learning in organizations. Applying the same type of analysis to artificial general intelligence (AGI) highlights some key theoretical difficulties and gives some indications about the challenges of resolving them.

**Keywords:** information; philosophy of information; artificial intelligence; machine learning; information quality; information friction

## 1. Introduction

The role of artificial intelligence (AI) and machine learning in organizations and society is of critical importance. From their role in the potential singularity (for example, see [1,2]) through their more pragmatic role in day-to-day life and businesses and on to deeper philosophical questions [3] they promise to make a widespread impact on our lives. Yet, on the other hand, they are just different forms of processing information.

However, information and information processing is beset with limitations that humans do not easily notice. As Kahneman [4] says with respect to our automatic responses (what he calls System 1): "System 1 is radically insensitive to both the quality and quantity of information that gives rise to impressions and intuitions." Yet information quality, what Kahneman says we are prone to ignore, is at the heart of many fundamental questions about information. Truth, meaning, and inference are expressed using information, so it is important to understand how the limitations apply. These topics are discussed in general in [5] and in [6–8] in respect of truth, meaning and inference more particularly.

In this paper, we take the same approach to AI and machine learning and consider the questions: how do the limitations and problems associated with information relate to AI and machine learning and how can an information-centric view help us to overcome the limitations? This analysis explains some current issues and indicates implementation principles required to resolve both pragmatic and deeper issues (A note on terminology: since machine learning is a subset of AI, where the context is broad, we will refer to AI and where the context is specifically about machine learning we will refer to machine learning).

The limitations of information arise from its evolution in information ecosystems in response to selection pressures [5] and the need to make tradeoffs to tackle the underlying combinatorial and pragmatic difficulties. Information ecosystems have different conventions for managing and processing

information. Think of the differences between mathematicians, banking systems and finance specialists, for example; each has their own ways of sharing information, often inaccessible to those outside the ecosystem. This approach to information is described in Section 2 that also describes the relationship of information with the interactions of Interacting Entities (IEs)—the entities, such as people, computer systems, organizations and animals that interact using information.

Following current ideas in technology architecture [9] and in usage traceable back to Darwin [10] we use the term fitness as a measure of how effectively an IE can achieve favorable outcomes in its environment. This interaction-led view leads to the following three levels of fitness that IEs may develop:

- Narrow fitness: that associated with a single interaction (and this is the type of fitness analyzed in [5–8]);
- Broad fitness: that associated with multiple interactions (of the same or different types) and the consequent need to manage and prioritize resources between the different types—this is the type of fitness linked to specialization, for example;
- Adaptiveness: that associated with environment change and the consequent need to adapt—this is the type of fitness that has led organizations to undertake digital transformation activities [11].

It is helpful to discuss fitness using some ideas developed for technology architecture [12]. Fitness needs a set of capabilities (where a capability is the ability to do something) that are provided by a set of physical components. Different components (e.g., web sites, enterprise applications, virtual assistants) are integrated together in component patterns (where the word "pattern" is used in the sense of the technology community [13]). Just as in technology architecture, these component patterns enable or constrain the different levels of fitness.

Using this approach, Section 3 builds on the analysis in [5–8] to highlight the limitations of information, how they apply to fitness in general, how they apply to AI and how AI can help to improve fitness. This section deals with current issues with machine learning and demonstrates a theoretical basis for implementation principles to:

- Understand the levels of fitness required and their relationship with information measures (like quality, friction, and pace [5,6]);
- Analyze the integration challenges of different AI approaches—the requirements for delivering reliable outcomes from a range of disparate components reflecting the conventions of different information ecosystems;
- Understand the best way to manage ecosystems boundaries—initially, how AI and people can work together but increasingly how AI can support effective interaction across other ecosystem boundaries;
- Provide assurance about the impact and risks as AI becomes more prevalent and the issues discussed above become more important to organizational success.

The theoretical difficulties become more profound when we consider artificial general intelligence (AGI) in Section 4. The following questions highlight important theoretical difficulties for which AGI research will require good answers:

- How is fitness for AGI determined?
- How will AGI handle the integration of components, the need to accommodate different ecosystem conventions and be sufficiently adaptive?
- How will AGI process and relate abstractions and will it be able to avoid the difficulties that humans have with the relationship between abstractions and information quality?

When we analyze these questions, it is clear that there are difficult information theoretic problems to be overcome on the route to the successful implementation of AGI.

## 2. Selection and Fitness

The relationship between information and ideas about evolution and ecology has been studied by several authors (see for example [14,15]). This section sets out the approach to information and evolution contained in [5–8]. In this approach, information corresponds to relationships between sets of physical properties encoded using conventions that evolve in information ecosystems. Consider the elements of this statement in turn.

Information processing entities interact with their environment, so we call them Interacting Entities (IEs—people, animals, organizations, parts of organizations, political parties, and computer systems are all IEs, for example). Through interaction, IEs gain access to resources such as money, food, drink, or votes for themselves or related IEs. Through a range of processes and feedback mechanisms, derived IEs (e.g., children, new product versions, changed organizations) are created from IEs. The health of an IE—its ability to continue to interact and achieve favorable outcomes—and the nature of any derived IE depend on the resources the IE has access to (either directly or through related IEs) and the outcomes it achieves. The interactions and outcomes available, together with the competition to achieve the outcomes, define the selection pressures for any IE. The selection pressures affect the characteristics of derived IEs. Selection, in this sense, is just the result of interactions. Examples of selection pressures include the market, natural selection, elections, personal choice, cultural norms in societies and sexual selection and for any IE different combinations of selection pressures may apply.

The ability of an IE to achieve a favorable outcome from an environment state requires information processing. For any environment state an IE needs to know how to respond, so it needs to connect environment states with potential outcomes and the actions required to help create the outcomes. Thus, IEs sense the values of properties in the environment, interpret them, make inferences, and create instructions to act. This information processing results in what is sometimes called descriptive, predictive and prescriptive information [7,8], corresponding to the categorization in Floridi [16] (Please note that these terms encompass other terms for types of information, such as "knowledge" and "intelligence").

The degree to which an IE can achieve favorable outcomes we call fitness, based on the extension of the Darwin's idea [10] in modern technology development [9]. There are three levels of fitness:

- narrow fitness: the ability to achieve favorable outcomes in a single interaction (this is discussed in detail in [5–8] including a discussion of the corresponding information measures: pace, friction, and quality);
- broad fitness: the ability to achieve favorable outcomes over multiple interactions, potentially of different types;
- adaptiveness: the ability to achieve favorable outcomes when the nature of interactions available in the environment changes.

Broad fitness takes into account factors that depend on multiple interactions. For example, there are many examples of machine learning in which human biases become evident over time [17,18]. These provide examples in which broad fitness can include ethical or social factors not always taken into account in narrow fitness or not evident in small numbers of interactions.

The degree of fitness depends on the component pattern of an IE. Here we are drawing on terminology used in IT architecture [12]. A component is a separable element of the IE—something that processes information in a particular way. In this sense, different applications and IT infrastructure are components for an organization; components for people are described in [19] (the authors say "inference, and cognition more generally, are achieved by a coalition of relatively autonomous modules that have evolved [ . . . ] to solve problems and exploit opportunities" and a "relatively autonomous module" corresponds to a component).

Figure 1 shows how these elements relate. In the figure, the superscripts 1, 2 and 3 refer to narrow fitness, broad fitness, and adaptiveness, respectively.

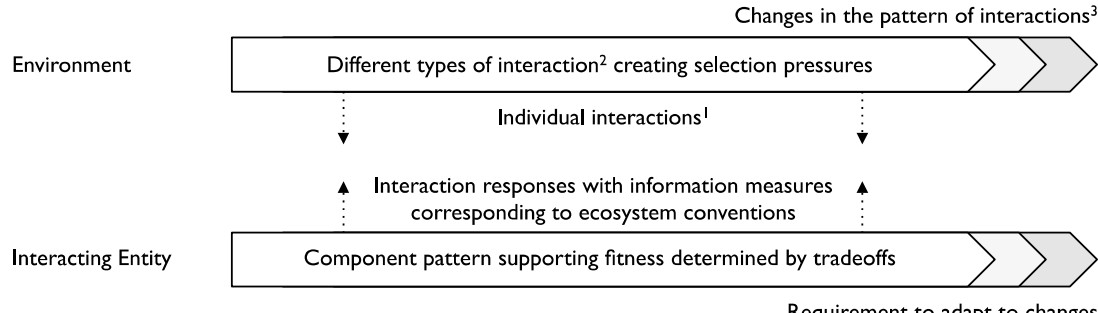

**Figure 1.** Levels of interaction and fitness.

Selection pressures lead to the formation of information ecosystems [5]. Examples include English speakers, computer systems that exchange specific types of banking information, mathematicians, finance specialists and many others. Each ecosystem has its own conventions for exchanging and processing information. Within different ecosystems, modelling tools (using the term from [5–8]) such as languages, mathematics and computer protocols have evolved to structure and manipulate information within the ecosystem. An IE outside the ecosystem may not be able to interpret the information—think of a classical languages scholar trying to understand quantum mechanics.

Information relates to the physical world. Call a slice a contiguous subset of space-time. A slice can correspond to an entity at a point in time (or more properly within a very short interval of time), a fixed piece of space over a fixed period of time or, much more generally, an event that moves through space and time. This definition allows great flexibility in discussing information. For example, slices are sufficiently general to support a common discussion of nouns, adjectives, and verbs, the past and the future.

Slices corresponding to ecosystem conventions for representing information we call content with respect to the ecosystem. Content is structured in terms of chunks and assertions. A chunk specifies a constraint on sets of slices (e.g., "John", "lives in Rome", "four-coloring"). An assertion hypothesizes a relationship between constraints (e.g., "John lives in Rome"). Within ecosystems and IEs, pieces of information are connected in an associative model (for example, Quine's "field of force whose boundary conditions are experience" [20], the World Wide Web, or Kahneman's "associative memory" [4]) with the nature of the connections determined by ecosystem conventions.

The effect of competition and selection pressures over time is to improve the ability of IEs and ecosystems to process information corresponding to different measures of information [5]. The quality of information may improve, in the sense that it is better able to support the achievement of favorable outcomes; it may be produced with lower friction [21] or it may be produced faster. Or there may be more general tradeoffs in which the balance between quality, friction and pace varies.

Selection pressures ensure that information is generally reliable enough for the purposes of the ecosystem within the envelope in which the selection pressures apply. However, quality issues and the limitations discussed below mean that outside this envelope we should not expect ecosystem conventions to deliver reliable results [6–8]. This is particularly important in an era of rapid change, such as the current digital revolution, in which IEs cannot keep pace with the change—for example creating the "digital divide" for people [22,23] and less market success for businesses [11]. For people, ecosystems can be age-related—for example, "digital natives", "digital immigrants" and "digital foreigners" [24] differ in their approach to the use of digital information.

### 2.1. AI and Machine Learning

AI is causing much debate at the moment. On the one hand it promises to revolutionize business [25] and on the other it may help to trigger the singularity [1,2,26]. The major recent developments in AI have been in machine learning—Domingos provides an overview in [27].

In this paper, we are concerned with the relationship between AI and information (as described in the previous section). As [25] demonstrates, AI can impact many elements of information processing for organizations. Importantly, it can make a significant improvement to all levels of fitness but to turn this into benefits, an implementation for an organization needs to link a detailed understanding of the three levels of fitness, their relationship and how each AI opportunity can improve them. In turn, this requires an understanding of measures of information such as friction, pace, and quality [5,6]. These points are expanded below.

## 2.2. Capability Requirements

To help understand how IEs can provide the levels of fitness required to thrive we can draw a capability model using a technique from enterprise architecture [12]. This approach is an elaboration of the approach taken in [5–8]. A capability is the ability to do something and we can draw a capability model for information capabilities, as in Figure 2, using the three levels of fitness identified in the previous section. Please note that this is a generic capability model that applies to all IEs and the degree to which capabilities are present in any IE may vary hugely. There are many other such models (for example, Figure 5 in [15]) highlighting different viewpoints but Figure 2 focuses on the issues that relate to fitness.

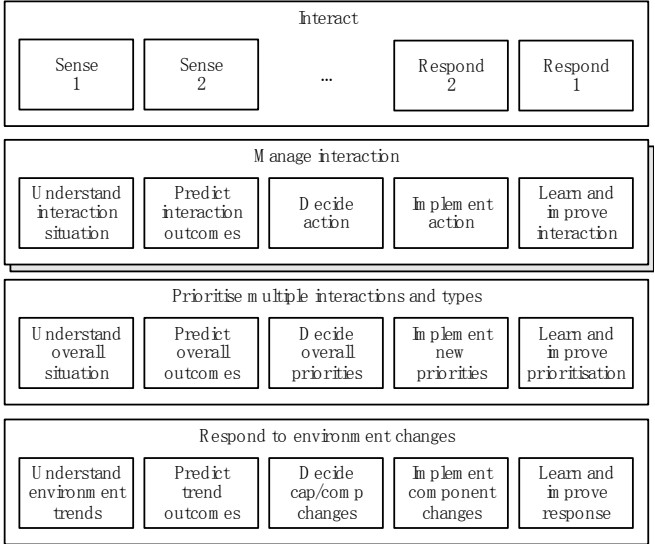

**Figure 2.** Information capability model.

An IE needs the capability to interact and, in turn, this needs the ability to sense and respond to the environment (for example, to understand speech and to talk). To manage the different levels of fitness it needs to be able to:

- manage each interaction and to decide how to respond (if at all)—each type of interaction may require different specific capabilities;
- prioritize the response to different interactions of the same or different types;
- respond to environment change—this is a priority for businesses as the world becomes increasingly digital and the implementation of AI and machine learning takes hold [11].

In each case there is a five steps process that applies at the appropriate level, involving:

- understand the situation—identifying what is relevant (distinguishing signal from noise), interpreting the relevant information in the environment by connecting it to information in memory, and distilling it to an appropriate level for analysis and decision-making;
- link the situation to potential outcomes and understand their relative favorability;

- decide how to respond;
- implement the change—in information terms this means converting the decision into instructional information;
- learn and improve.

Each capability describes what an IE could do but not how it does it or the degree to which it does it. Any particular IE will have a set of components conforming to a component pattern that provides the capabilities. The nature of component patterns is discussed in Section 3.

## 3. The Limitations of Information and Applications of AI to Business

Information and its processing is subject to many limitations—these are discussed at length in [5–8]. These limitations occur because it is difficult linking environment states with future outcomes and the required actions to achieve favorable outcomes under the influence of selection pressures. The impact is always that perfection is impossible and under different selection pressures there are tradeoffs with respect to the different components of fitness.

This section provides an overview of these problems and how different ecosystem conventions and modelling tools can help to overcome them. In particular, it discusses the impact on the problems of AI and how the use of AI can, in some cases, help to resolve them.

The first problem is combinatorial. The number of possible environment states, outcomes, and the relationships between them is huge and in each interaction, these must be boiled down by an IE to a single action (including the possibility of inaction). The basis for overcoming this problem is provided fundamental characteristics of information—symbols and the means of associating them in different ways.

The second problem is how to make the tradeoffs between the information measures (like pace, friction, and quality [5,6]) required to support favorable outcomes. This problem breaks down into several sub-problems:

- what properties to measure in the environment and to what quality?
- how to overcome the complexities involved in interpretation, inference, and instruction—how to develop shortcuts in the provision of the required quality?
- how to overcome contention at different levels—between different information measures, between the needs of the present and the needs of the future, between the needs of different interaction types, between different ecosystems (especially at ecosystem boundaries) and how to keep the IE aligned with fitness even as the requirements of the environment change?
- how to challenge the level of quality achieved—how the ecosystem can apply selection pressures of its own to ensure quality?

The final problem is architectural (in the sense of [12])—what component pattern is best and how should this change over time?

The ways in which these problems are resolved in different ecosystems determine the ecosystem conventions and the detailed selection mechanisms that apply.

### 3.1. The Combinatorial Problem

Information overload has been much discussed [28] but this is but one symptom of a deeper problem: there is an unimaginably large number of measurable potential environment states, potential outcomes, and connections between them.

An environment state or, indeed, any slice, even if measured with relatively poor quality, is not easy to manipulate and process—there are a (potentially large) number of properties and their values to consider. Therefore, there is a large processing saving (and reduction in friction and pace) if a simple identifier, associated with the slice, is used instead. If the identifier is connected, in some way, with the slice and it is clear what the identifier means (by reference to the slice properties, as needed), then

processing will be simplified hugely. Therefore, it is unsurprising that identifiers are widespread in information storage and processing in the form of symbols or sets of symbols. The nature of the symbol is not relevant (and the fact that symbols can be arbitrary is a fundamental principle of semiotics [29]). What matters is that the symbol can be connected, as needed, to the slice it is connected to and that it can be discriminated from other symbols. (Please note that the requirement that symbols can easily be discriminated foreshadows one of the benefits of the digital world—see the discussion in [30].)

This helps to solve the processing problem but if we need a symbol for each possible slice then we have not escaped the combinatorial problem entirely. It would also be useful for a symbol to apply to a set of slices that have something in common—that meet some set of constraints. This is, for example, the way language works: verbs relate to sets of event slices with common properties; adjectives relate to sets of slices with some common properties and so forth.

Set inclusion is binary: in or out. Therefore, by taking this route to solve the combinatorial problem, the use of symbols has built in a fundamental issue with information that underpins many of the limitations analyzed in [5–8]. The authors discuss this question in their analysis of patterns in [28] and say: "It is paradoxical that the similarity of the elements of a set creates a difference between the very elements of the set and all of the things not in the set". If one or more pieces of content maps close to the boundary of a set (in the sense that a small change in property values moves it to the other side of a boundary) then an interpretation, inference or instruction that relies on that positioning requires the quality to be high enough to guarantee the positioning. Call this the discrimination problem. An extreme form of the discrimination problem arises from chaotic effects [31] in which arbitrarily small changes can give rise to large outcomes. As demonstrated in [5,6], much routine information processing ignores this question entirely. In machine learning terms, the discrimination problem translates into the levels of risk and tolerance associated with false positives and false negatives [32].

The use of symbols enables another trick: symbols can relate to other symbols not just to sets of slices (this is because symbols correspond to sets of slices conforming to constraints in a particular ecosystem [5]). Therefore, as described in [6,7], we need to be careful to distinguish between content slices—those interpreted as symbols in an ecosystem (by IEs in the ecosystem)—and event slices—those that do not.

Of course, all the discussion about symbols is ecosystem-specific. A symbol in one ecosystem may not be one in another—words in one language may not be in another language, mathematics is meaningless to non-mathematicians.

The combinatorial challenge is magnified when we consider multiple interaction types and environment change. Multiple interaction types may need more slice properties, more symbols and, perhaps, different ecosystems and ecosystem conventions. In addition, recognizing environment change requires the ability to store and process historical data that will allow the identification of trends (access to this historical "big data" has been one of the drivers of machine learning).

This leads to another aspect of the combinatorial challenge: how should information and components be structured to enable fitness at the various levels (including adaptiveness). Remembering that information is about connecting states, outcomes and actions, there is a key structuring principle here (used commonly in the technology industry [9]). Decoupling two components enables one to be changed without changing the other (decoupling is discussed more in the discussion about component patterns below) and this requires them to be separable in some sense. We can replay the discussion above in the following way:

- The use of symbols separates information from source slices and their properties;
- Ecosystem conventions separate symbols from particular slice representations (so words can be written or spoken, for example);
- Evolving ecosystem conventions separate processing (and the making of connections) from particular IEs (so computers can automate some human activities, for example);
- Communication separates content from a physical location (so content can be duplicated at distance).

In this way, the evolution of ecosystem conventions progressively frees up information from the particular process that generates it. This progression is neatly reflected in the development of organizational enterprise architectures [12] in which two major themes have emerged:

- Developments in data warehouses, business intelligence, data engineering, data lakes and data hubs enable the collation and manipulation of data from many different sources;
- Digital technologies enable information to be available at widely different times and places and on many different devices.

One strategy for addressing the combinatorial problem is increasing processing power and this is precisely what Moore's law [33] has provided for machine learning (combined with access to access to large volumes of data—so-called "big data"). This increase in power and access to data has been one of the drivers of the current boom in AI but is, as yet, a considerable distance away from resolving the combinatorial problem, even aside from the other difficulties outlined below.

### 3.2. Selection Tradeoffs, Viewpoints and Rules

The impact of the combinatorial problem is that information processing uses a strict subset of the properties of environment states available, makes quality tradeoffs and may be linked to a strict subset of possible outcomes. In other words, all information processing has a viewpoint (using the terminology employed in [7,8]). This is routine in day-to-day life—for example:

- with the same evidence, different political parties reach very different conclusions about the right course of action in any case;
- in legal cases, the prosecution and defense represent different viewpoints;
- even in science, there are divisive debates about the merit of particular hypotheses (this is represented, for example, in Kuhn's philosophy of science [34]).

Since these viewpoints are inevitable, we need to understand their impact. This is the focus of the following sections.

### 3.2.1. Measurement

Measurement is about converting environment states into properties and values or more abstract content (subject, of course, to the prevailing ecosystem conventions). How does this relate to fitness measures?

One dimension is the number of properties measured, how they are measured and the quality of the measurement. In addition, once properties are measured, how often do they need to be re-measured—to what extent is timeliness an issue [5]?

When multiple types of interaction are considered, an extra dimension comes into play—to what extent can measurement required for one interaction type be used for another—if the different interactions use different ecosystem conventions, can the properties be measured and processed in the same way and what are the implications if they are not? This a common problem in organizations—the quality of information needed to complete a process successfully may be far less than that required for accurate reporting.

Finally, when the environment is changing, there may be a requirement for new properties to be measured or for changed ecosystem conventions to be considered.

Machine learning can be one of the drivers behind improved measurement for organizations because the recognition of patterns and its automation [27] are fundamental principles in the discipline [32]. Machine learning can improve pace, reduce friction and, in some cases, improve quality also through the automation of learning based on good quality data (although there have been some significant difficulties [17,18]).

3.2.2. Information Processing Limitations and Rules

As discussed in [5–8], different strategies are possible for information processing depending on the degree to which each of quality, pace or friction is prioritized in terms of narrow fitness. A rigorous process focusing on quality requires an approach such as that of science but many ecosystems cannot afford this overhead. Instead they rely on rules that exploit the regularities in the environment, as discussed by the authors in [19], who say:

*"What makes relevant inferences possible [ … ] is the existence in the world of dependable regularities. Some, like the laws of physics, are quite general. Others, like the bell-food regularity in Pavlov's lab, are quite transient and local. [ … ] No regularities, no inference. No inference, no action."*

There can be difficulties associated with exploiting these regularities both for people and machines. As Kahneman points out [4] with respect to our innate, subconscious responses (what he calls System 1): "System 1 is radically insensitive to both the quality and quantity of information that gives rise to impressions and intuitions." As Duffy says in [35] "and the more common a problem is, the more likely we are to accept it as the norm".

Machine learning [27] finds and exploits some of these regularities but has been subject to some well-publicized issues associated with bias [17,18] (although the biases revealed have, in some cases, been less than people display [18]).

The nature of the regularities is discussed in [8] in which inference is categorized in terms of:

- Content inference—using only the rules associated with a particular modelling tool (for example, formal logic or mathematics);
- Causation—in which inference is based on one or more causation processes;
- Similarity—in which inference is based on the similarity between sets of slices and the assumption that the similarity will extend.

Machine learning is based on similarity, so this categorization poses a question. For what types of information processing is machine learning the most appropriate technique and when are other techniques appropriate? In particular, when is simulation (concerned with modelling causation) more appropriate? This question is discussed in Section 4.

Content processing has clear benefits in terms of friction and pace—making the connection with events incurs much higher friction (this is the relationship between theoretical physics and experimental physics, for example, and consider the cost of the Large Hadron Collider). Wittgenstein also referred to this idea and the relationship between content and events [36,37] with respect to mathematics:

"[I]t is essential to mathematics that its signs are also employed in mufti";

"[I]t is the use outside mathematics, and so the meaning ['Bedeutung'] of the signs, that makes the sign-game into mathematics".

An equally insidious shortcut is output collapse (to use the term used in [8]). There are uncertainties about interpretation, inference and instruction caused by information quality limitations. However, an interaction results in a single action by an IE (where this includes the possibility of no action at all) and examining a range of potential outcomes and actions increases friction. Therefore, in many cases, interpretation and inference are designed to produce a single answer and the potentially complex distribution of possibilities collapses to a single output. If this collapse occurs at the end of the processing, then it may not prejudice quality. However, if it occurs at several stages during the processing then it is likely to.

There is another type of shortcut. This is quality by proxy in which quality is assessed according to the source of the information (linked to authority, brand, reputation, conformance to a data model or other characteristics). In [38], the authors express this idea elegantly with respect to documents: "For information has trouble, as we all do, testifying on its own behalf... Piling up information from the same source does not increase reliability. In general, people look beyond information to triangulate reliability."

As a result, of selection tradeoffs, these various types of shortcut become embodied in processing rules that are intended to simplify processing with sufficient levels of quality. The rules are defined with a degree of rigor consistent with ecosystem conventions (for example, rigorous for computer systems but less so for social interaction).

Organizations use rules such as this (called business rules) routinely. Business processes embody these business rules in two senses. At a large scale, a process defines the rules by which a business intends to carry out an activity (for example, how to manage an insurance claim). In addition, in a more detailed sense, business rules capture how to accomplish particular steps (for example, the questions to ask about the nature of the claim). Machine learning can improve both of these aspects. In the first case, the context of the process (for example, information about the claim) may change the appropriate next step (for example, the appropriate level of risk assessment to apply). Therefore, rather than a fixed set of steps as captured in a process map, the process may become a mixture of fixed steps and something akin to a state machine [39] or, in some cases, just a state machine. This change relies on a continuous situation awareness (as described in Figure 2) that can use machine learning as a measurement tool. In addition, machine learning can also refine the business rules over time based on the developing relationship between the rules and fitness objectives (for example, the tradeoff between quality and friction or pace). It may be appropriate to change the rules (changing the questions to ask in this example) when more information is learnt about the effectiveness of the rules or it becomes possible to tune the rules more specifically to individual examples.

### 3.2.3. Contention

Selection tradeoffs are about managing contention and ecosystem conventions embed the tradeoffs. For a single interaction there is contention between pace, friction, and quality. This type of contention is discussed in detail in [5–8].

Multiple interactions and types of interaction introduce extra dimensions. The first is between the present and the future: how much should an IE optimize the chances of a favorable outcome for a single interaction against the possibilities of favorable interactions in the future? The second is between different interaction types: how much should an IE focus on one type of interaction compared to others? Or, to put it another way, how much should the IE specialize? Many authors in different disciplines have discussed specialization as a natural outcome of selection pressures—for example:

- Philosophers from Plato [40] onwards discussing the division of labor;
- Biologists including Darwin [10], since species themselves are examples of specialization;
- Business writers discussing differentiation, including Porter [41].

More generally, there might be what we can call conflict of interest between narrow fitness and broad fitness especially when the nature of quality associated with narrow fitness does not match that associated with broad fitness. In [42], the author gives examples of the impact of conflict of interest on science. There have been several well-publicized examples concerning machine learning [18]. In these cases, narrow fitness is defined in terms of the data used to generate the learnt behavior but the data itself may embed human biases. As a result, narrow fitness (linked to training data) does not take ethical and social issues into account and broad fitness is reduced.

The next point of contention arises from ecosystem boundaries. The conventions that apply on one side of the boundary may be very different from the other (we only need to consider speakers of different languages or the user experience associated with poorly defined web sites) and there may be contention at fundamental levels. One initial driver of AI (the Turing Test [43]) was aimed at testing the human/computer ecosystem boundary. This is still of considerable importance but a related question in organizations is understanding how AI and people can work together [44] and how AI can support other ecosystem boundaries.

Finally, there may be contention in the balance of the selection pressures as the environment changes. For example, in the digital revolution engulfing the world of business [11] the balance

between friction, pace and quality is changing—the ability to respond fast (i.e., pace) is becoming more important. Machine learning plays a part here since it is a mechanism for constantly re-learning from the environment.

### 3.2.4. Challenge and Assurance

For an IE, information processing is reliable if it helps to achieve a favorable-enough outcome—if the IE can rely on the processing within the envelope provided the ecosystem selection pressures (as discussed in [6–8], outside this envelope is it not guaranteed to be reliable enough). Therefore, how can ecosystems apply their own selection pressures to improve the reliability of information processing? An element that many ecosystems have in common is that of challenge. Table 1 (copied from [8]) shows some examples.

**Table 1.** Challenge.

| Ecosystem | Hypothesis | Challenge |
|---|---|---|
| English criminal law (prosecution) | The defendant is guilty | The defense (plus, potentially, the appeals process) |
| Science | A prediction made by a hypothesis is true | Experiments to refute or confirm the prediction |
| Mathematics | A theorem is proved | Peer review |
| Computer systems | The system will perform as required | Tests that the system meets its requirements |

The objective of each challenge is to identify weaknesses in information processing either in terms of its output (e.g., refutation in scientific experiments), the input assertions on which it is based (e.g., the evidence in a trial) or the steps of the inference (e.g., peer review in mathematics).

The generic mechanism is similar in each case. A related ecosystem has selection pressures in which favorable outcomes correspond to successful challenges. The degree to which the challenge is rigorous depends on the selection pressures that apply to it and, in some cases, the degree to which a different IE from the one making the inference conducts it (to avoid the conflicts of interest discussed in [42], for example).

Therefore, given that challenge is a type of selection pressure, how does the nature of challenge relate to fitness criteria? There are some obvious questions. First, is the inference transparent enough to be amenable to challenge? This is one of the questions that has been raised about deep learning although recent research has started to address this question [45].

Secondly, what is the degree of challenge—how thorough is it? This is an important issue addressed by organizations as they implement machine learning—how does the assurance of machine learning relate to conventional testing and are additional organizational functions required. This is discussed below.

Thirdly, what is the scope of the challenge is relation to fitness—is it concerned with narrow fitness or does it incorporate broad fitness and adaptiveness as well? This is one of the considerations described in detail with respect to technology in [9]; but the issue as applied to machine learning is more extensive because machine learning learns from historic data that may not encapsulate the desired requirements of broad fitness and is unlikely to include the requirements of adaptiveness.

Challenge and assurance is important for machine learning since there are many public examples in which machine learning has delivered unacceptable results [17,18]. An element of broad fitness that has been the subject of much attention is ethics [46], because of these issues and also the long-term direction of AI and the potential singularity [1,2,26].

The purpose of the challenge is to identify what the software industry calls test cases [39]—a set of inputs and outputs designed to cover the range of possibilities thoroughly enough to provide confidence of reliability (in the context of the ecosystem conventions). In clearly defined domains such as Go and chess, the test cases themselves can be generated by machine learning but where there

is a level of organizational risk involved (e.g., reputational, ethical, operational or security-related) then more traditional forms of assurance may be required focusing on the training data, the selection of a range of scenarios to test and an organizational assurance function to analyze examples of the discrimination problem and potential impacts. Since machine learning can re-learn periodically, these forms of assurance may need to be applied, in some form, regularly.

Therefore, we can conclude that, as AI becomes more prevalent and the issues discussed above become more important, organizations will need to understand and manage the potential impacts and risks. This will require an organizational assurance function that will ensure that the right degree of challenge is applied and analyze and, where necessary, forecast the impact of AI on business results.

### 3.3. Component Pattern

Components are the physical realization of capabilities (see Figure 2) and components can be arranged in different patterns. Table 2 shows some examples of components. The relationship between capabilities and components for business and technology architectures is part of the day-to-day practice for enterprise architects [12]. The development of component patterns to meet future fitness requirements is a key part of developing future architectures to support organizational fitness requirements [9]. We can use these ideas to analyze component patterns for IEs.

**Table 2.** Examples of components.

| IE | Interaction Component | Interpretation, Inference, and Instruction Component |
|---|---|---|
| People | Senses (eyes, . . . ) | Different brain mechanisms (see [19]) |
| Organizations | Sales people, customer research, web sites, . . . | Different organizational functions and their supporting computer systems (for example, qualifying sales opportunities, deciding the chances of winning and deciding how to price the product or service) |
| Computer system architectures | Virtual assistants (e.g., Alexa, Siri), apps, enterprise applications, security intrusion detection, . . . | Algorithms, machine learning tools |

Components evolve incrementally and become integrated to meet the need to connect environment states to outcomes and actions. The nature of the integration and the pre-dominance of certain components can imply different patterns. These patterns have a set of characteristics based on the capabilities shown in Figure 2:

- Channel-aligned: in this case, interaction components extend to encompass wider information processing. For example, Pinker [47] gives many examples in which the processing of the human brain is influenced (and constrained) by language and, indeed, some believe that language processing underpins all of human thought (for example in [48] the author says "I believe that language is also the medium by which we formulate our conceptual thinking. I regard thinking as silent language.").

- Function-aligned: in this case, components (like interaction components) are built out from particular functions. For example, many organizational capabilities (such as finance and HR) are supported by software products that have developed from a functional base and also provide interaction (e.g., through web sites) and analytics.

- Multi-function: in this case, different components providing different functions are integrated together. For example, in [19], the authors make the case that many specialized inference mechanisms have evolved in people; they say: "inference, and cognition more generally, are achieved by a coalition of relatively autonomous modules that have evolved [ . . . ] to solve problems and exploit opportunities". Another example is an extension of the previous case in which organizations have specific components to support finance, HR, manufacturing, retailing and other organizational functions.

- Information-aligned: in this case, components are based on the capability model in Figure 3. For example, many organizations have built data warehouses to support business intelligence as well as data lakes and analytics capabilities [49].

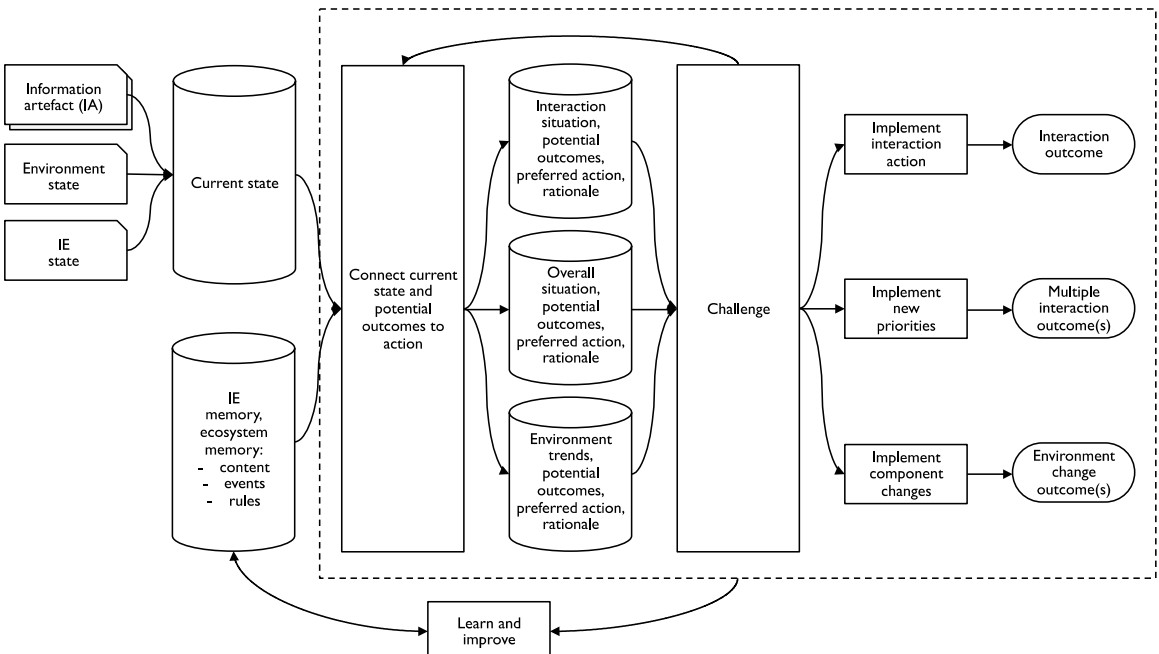

**Figure 3.** Information-aligned component pattern.

These different types of pattern have different strengths and weaknesses based on the core components. IEs often have a combination of these patterns and the balance between them impacts elements of fitness. Component patterns embed the information structure and processing tradeoffs implicit in ecosystem conventions and these both enable and constrain different elements of fitness. For example, channel-aligned patterns are strong when interaction is a large element of fitness; information-aligned patterns are strong when information needs to be integrated separately outside the processes that generated the information.

However, component patterns may need to change. For example, there is a clear trend [30] for organizations to respond to the digital economy by adding an information-aligned pattern that takes advantage of machine learning. Figure 3 shows a generic information-aligned component pattern.

A more extreme example of change is the trend towards the AI-assisted human and the need for humans and AI to work together [44].

Components need to be integrated in order to link environment states to outcomes and actions. Narrow fitness demands short and efficient processing embedding rules that deliver sufficient quality. Broad fitness requires additional processing complexity and may also require the integration of different ecosystems with different conventions. Both of these are drivers for tight integration between components.

However, adaptiveness requires decoupling—the ability to change components independently [9]—because otherwise change incurs too much friction. This generates a tension between the different types of fitness; without a sufficiently strong adaptiveness selection pressure, the nature of the component integration can be brittle and resist change.

For organizations, machine learning has a role to play here. If some or all of the business rules are based on machine learning, then periodic re-learning can update the rules (but see the discussion about re-learning below). For this to be the case, the organization will need a component pattern that is sufficiently information-aligned. As AI becomes embedded in more and more technology, the

shift towards information-alignment, or the addition of information-alignment, will become more and more important.

The same change (towards information-alignment) is also true of quality improvement. Better-informed people make better decisions and the same principle underpins the implementation of machine learning in business. Improvements in interpretation and inference quality require richer access to information [5,8] that channel-alignment or function-alignment alone cannot provide.

In [19], the authors demonstrate that human inference has many different inference patterns. In addition, the mind does a wonderful job of giving us the illusion that things are well integrated even when, underneath, they are not; this is what magic relies on [50]. Machine learning may be heading in the same direction—current developments in AI contain several different patterns. Domingo [27] categorizes these as symbolists, connectionists, evolutionaries, Bayesians, and analogizers. However, more generally, AI is becoming a set of techniques embedded in numerous applications using whatever technique(s) is appropriate in each case. In this case, the integration question takes on another dimension: how can the interpretations and inferences of multiple components including AI integrate into reliable interpretations and inferences for the organization as a whole. The different components may use different ecosystem conventions with different information structures and process tradeoffs. There may be gaps between their domains (as in the magic example above). Other problems identified above (output collapse and contention) may apply. There is also an uneasy relationship between AI component integration and the discrimination problem. If inference relating to a critical boundary condition relies on integration between machine learning components, then the reliability of the integration needs to be tested rigorously.

The challenge becomes greater when we consider re-learning. One of the advantages of machine learning is that rules can be re-learnt as the environment changes. However, when many machine learning components are integrated to support a complex set of business functions, how should this re-learning work? Again, the principle of decoupling applies—we want the different interpretations and inferences to be independent. However, how do we know that this is the case? With more data or a change in the environment, new patterns may emerge in the data (that, after all, is the whole point of re-learning) and these new patterns may create new dependencies between the rules. This reinforces the need for assurance (as discussed above).

Therefore, we can conclude that, as machine learning becomes more pervasive, integrating different approaches to machine learning, each supporting different viewpoints and ecosystem conventions, will provide challenges in the following four areas:

- Providing high quality, coherent descriptive, predictive, and prescriptive information from disparate components each learning in different ways from different subsets of data at different times;
- Tackling the discrimination problem especially where components need to be integrated;
- Ensuring that content information processing does not suffer from the same limitations as for humans;
- Ensuring that the underlying data is of the required quality for each component.

These challenges provide a foretaste of the deeper issues with AGI discussed in the next section.

## 4. The Limitations of Information and AGI

AGI is one of the main factors driving AI research (see, for example, [26]) and, in the view of many authors (for example, [1,2]), AGI is a step on the road to the singularity. Therefore, it is important to understand the impact of the limitations of information and the theoretical and practical difficulties that they imply about AGI.

In this section, we discuss the following challenges for AGI based on the analysis above:

- How is fitness for AGI determined?

- How will AGI handle the integration of components, the need to accommodate different ecosystem conventions and be sufficiently adaptive?
- How will AGI process and relate abstractions and will it be able to avoid the difficulties that humans have with the relationship between abstractions and information quality?

One difference between narrow AI and AGI is that AGI needs to handle many interaction types and combinations of them, so how is it possible to define or characterize all of them? And how can we apply the right selection pressures—to use the terminology of IT, how can we define all of the test cases required? One approach of the AI community is to use AI techniques (like adversarial generative networks) to this further question. However, for difficult questions, and for broad fitness in general, at some stage people will need to be sure of the potential outcomes, so people will need to apply the right selection criteria even to those further AI techniques. It is difficult to see this as other than another manifestation of the combinatorial problem but magnified by the number of different interactions types and their combinations. Defining broad fitness for people and organizations includes the legislation of a country as well as cultural and moral imperatives, so how can we define it for AGI? (This topic has been recognized widely including by such multi-national bodies as the World Economic Forum who ask the question "How do we build an ethical framework for the Fourth Industrial Revolution" [51]). As well as these aspects, broad fitness for AGI will require rigorous security fitness. The combination of all of these is a dauntingly large task.

This implies that it is very difficult to define even what AGI is in enough detail to be useful in practice. In addition, we need a specific definition because overcoming the discrimination problem requires appropriately high information quality—for AGI, the discrimination required may include many issues concerning human safety, as we have already seen with autonomous cars.

One way round this is the AGI equivalent of "learning on the job"—allowing AGI to make mistakes and learn from them in the real world. Whether or not this is feasible depends on the fitness criteria that apply—it is difficult to see that this would be acceptable for activities with significant levels of risk. It has already caused reputational damage in the case of simple, narrow AI [17,18]. In [52], the authors address this issue when they ask the question: "why not give AGI human experience"? They then show how human experience is difficult to achieve. Given the discussion in Section 3 about viewpoints, if the experience of AGI is different from human experience then, necessarily, its viewpoint will be different and its behavior will be correspondingly different.

How about integration? In humans, different types of interpretation and inference use different components [19]. Currently, the same is true of machine learning—increasingly, it is a computing technique that is applied as needed. Therefore, it seems likely that AGI will need to integrate many different learning components. Domingos [27] suggests one integration approach and there are other approaches (e.g., NARS [53,54]). There are several issues here: ecosystem conventions, content inference, selection tradeoffs and component patterns.

Just as people may engage with different ecosystems (e.g., different languages, different organizational functions, computer systems, different fields of human endeavor (sciences, humanities)) AGI will need to be able to deal with different ecosystems and their relationships. Different ecosystems have different conventions and fitness criteria so AGI will need to manage these and convert between them. Again, the discrimination problem raises its head—different ecosystem conventions are not semantically interoperable. Combining processing using different ecosystem conventions risks what [7,8] refer to as "interpretation tangling" or "inference tangling" in which conventions that apply to one ecosystem (e.g., mathematics) are implicitly assumed to apply to another (e.g., language) resulting in unreliable results. A learning approach could only address these issues if the combinatorial problem described above does not apply (and in reality, it may not be possible even to identify or source all the possible combinations to learn).

Deep learning uses layers of neural networks in which intermediate layers establish some intermediate property and subsequent layers use these abstractions; thus, these subsequent layers are then using content processing. Metalearning [27] provides another example of content processing.

In these examples, because they apply to narrow AI, the limitations of content processing described in Section 3 have little impact. However, when we scale up to AGI with many components of different types developed for different ecosystems providing abstractions that are integrated by one or more higher levels of machine learning then the limitations of content processing may become a problem.

Content processing is used by ecosystems because the use of content rules is much faster and more efficient than event processing (testing against the properties and values of sets of slices)—this is an outcome of the combinatorial problem. Therefore, is it feasible that this requirement not be present for AGI? Only if the AGI could relate all information processing to events (not content) as it was needed. In the face of the discrimination problem this amounts to the ability to provide processing power to overcome much of the combinatorial problem. Even if Moore's law [33] continues, this is a difficult proposition to accept for the foreseeable future and even if it was feasible, there is no guarantee that it would not be subject to selection tradeoffs.

Therefore, we can conclude that content processing will likely be a part of AGI and therefore that the limitations of content processing will also apply and that, as a result, information quality will be compromised. However, without a definite AGI model to base the analysis on, the impact of this is unclear.

What about adaptiveness? Adaptiveness is, partly at least, an attribute of the component pattern. However, the experience from the technology industry, most recently in developing digital enterprise architectures [55] is that developing new component patterns is a change of kind not of degree—component pattern changes are difficult to evolve by small degrees. Thus, we cannot expect linear progress. This is discussed in [52] in which the authors include the following quote from [56] "The learning of meta-level knowledge and skills cannot be properly handled by the existing machine learning techniques, which are designed for object-level tasks". Perhaps AGI will need the ability to learn about component patterns themselves—when a new component pattern is needed the AGI will need to recognize it and evolve a new one; but even if this is feasible, where will the data come from?

In principle, AGI could be adaptive, within the context of a single component pattern, because it can re-learn periodically. However, re-learning will be subject to selection pressures and the possibility of tradeoffs and different ecosystem conventions. Thus, in practice, different machine learning components may re-learn at different rates and times raising the possibility of inaccuracies and inconsistencies exacerbating the discrimination problem and quality in general.

As Section 3 points out, the degree of decoupling within the component pattern is important for adaptiveness. The human brain masks the cognitive integration difficulties we all have [50] between different components. It is possible that this type of integration difficulty is a natural consequence of the tradeoffs between adaptiveness and other levels of fitness. Can we be sure that the same does not apply to AGI?

The discussion about information processing in Section 3 (and [8]) highlights another potential difficulty with machine learning and AGI. One of the prevalent ideas in technology at the moment, driven partly by the Internet of Things and the ability to understand the status of entities, is that of the "digital twin"—a simulation of those entities. Similar ideas are driving technologies such as virtual reality and, of course, in many scientific and other fields, simulation has long been a critical tool. Bringing these ideas together will support the creation of models of the environment enabling a richer simulation of external activities, leading to the question: under what circumstances will simulation be preferable to AI and how can they work together?

Machine learning exploits "the existence in the world of dependable regularities". However, will these dependable regularities occur reliably enough in the information available to machine learning to provide sufficient quality to overcome the discrimination problem? Might not inference based on causation be required to address some difficult instances of the discrimination problem? This question is the AI equivalent of the "blank slate" issue discussed by Chomsky [57] and many others. Since complex simulation relies on complex theoretical models, inference based on causation it is not, in the foreseeable future, amenable to machine learning.

## 5. Conclusions

The analysis of fitness and the limitations of information above provide a sound theoretical basis for analyzing AI both for implementation in organizations now and with respect to AGI. This analysis is validated by the current experience of AI and can also be used to define the following important implementation principles.

- Fitness: AI can make a significant improvement to all levels of fitness but to turn this into benefits the implementation of AI for organizations should be based on a detailed understanding of the three levels of fitness, the relationship of the levels and how each AI opportunity can improve them. In turn, this requires an understanding of measures of information such as friction, pace, and quality.
- Integration: Organizations will need to analyze the integration challenges of different AI approaches. As AI becomes more pervasive, integration will provide challenges in the following four areas:

  ○ Providing high quality, coherent descriptive, predictive, and prescriptive information from disparate components each learning from different subsets of data at different times using different techniques;
  ○ Tackling the discrimination problem especially where components need to be integrated;
  ○ Ensuring that content processing does not suffer from the same limitations that it has for humans;
  ○ Ensuring that the underlying data is of the required quality for each component.

- Ecosystem boundaries: One initial driver of AI (the Turing Test) was aimed at the human/computer ecosystem boundary. This is still important but a related question in business is understanding how AI and people can work together and how AI can support other ecosystem boundaries.
- Assurance: As AI becomes more prevalent and the issues discussed above become more important, organizations will need to understand and manage the potential impacts and risks. This requires an organizational assurance function that will analyze and, where necessary, forecast the impact of AI on business results.

These topics increase in importance with respect to AGI because the theoretical difficulties will become more profound. The following questions highlight important theoretical difficulties for which AGI research will require good answers:

- How is fitness for AGI determined?
- How will AGI handle the integration of components, the need to accommodate different ecosystem conventions and be sufficiently adaptive?
- How will AGI process and relate abstractions and will it be able to avoid the difficulties that humans have with the relationship between abstractions and information quality?

When we analyze these questions, it is clear that there are difficult information theoretic problems to be overcome on the route to the successful implementation of AGI.

**Funding:** This research received no external funding.

**Conflicts of Interest:** The author declares no conflict of interest.

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
