# Peer review of "Artificial Intelligence and the Limitations of Information"

_information, doi:10.3390/info9120332_

Round 1

Reviewer 1 Report

I am not a philosopher, so perhaps I just can't appreciate this kind of paper.  The final concluding sentence is "

"When we analyse these questions it is difficult to avoid the conclusion that the route to AGI is

not just a linear extrapolation of current technology capabilities and that there are major difficulties to be overcome."

This seems obvious, there was no need to go through all the typologies and definitions of terms just to get to this.  If the author could give one example of how some of these concepts help to understand some of the problems that have emerged in developing AGI, I could see some sense in plowing through all the abstractions. 

I don't have any criticisms of any of the typologies, they just don't interest me and I don't see their value.  If other reviewers, perhaps with a different disciplinary background, find them of interest I wouldn't object to them being published.  When I say it needs a major revision, I mean it needs at least one empirical example.

Author Response

The paper is about the relationship between information theory, organisational practice and AI—it is about challenges in implementing AI/AGI successfully not just about AI/AGI in a research context. It shows that the analysis of fitness and the limitations of information provide a sound theoretical basis for analysing AI both for implementation in organisations now and with respect to AGI. 

The conclusion of the paper is about the specific information theory ideas that are discussed in the paper. The concluding sentence (as quoted by the referee) references some specific questions in the previous paragraph that show the link with the information theory ideas analysed. To clarify the relationship with these questions and the preceding analysis in the paper, I have changed the concluding sentence at the end (and in the introduction) to “When we analyse these questions it is clear that there are difficult information theoretic problems to be overcome on the route to the successful implementation of AGI.” The root causes of several current implementation difficulties lie with the information theoretic ideas discussed in the paper. The fact that even narrow AI encounters these difficulties means that the information theoretic ideas will need to be specifically addressed with AGI. 

The referee asks for an example. The paper highlights the relationship of AI implementation with business processes, business rules, ethics, conflict of interest, assurance, technology architecture and digital transformation, including some examples. 

Reviewer 2 Report

This is a very well written paper, alas based on a philosophical perspective that I personally do not find correct.

As a physicalist/naturalist/etc. I cannot fully appreciate line of thoughts like the one presented by the author, which gives information such a primitive and fundamental role.

Many share the same hypotheses and end up in the most absurdly unfounded sci-fi.

Luckily, the author’s feet stay on the ground throughout the work, which only indirectly hints at a fantastic theory of everything revolving around information, without becoming one.

The conceptual links with AI are a bit weak, because some considerations are so general, they may apply to computer science in general, but the references to reasoning and causation suffice in showing that the focus is indeed on this discipline.

There are no research results. This paper is to be considered as an extended opinion paper with a significant overview on the field.

I am not at all convinced about this conceptual approach, but it is very clearly explained and illustrated. 

Author Response

I don’t understand the referee’s reference to sci-fi. In my reading of the paper there is nothing that relates to sci-fi and nor is there any intention to relate to sci-fi. Indeed, the analysis in the paper focuses on the pragmatic difficulties of implementing AI/AGI. 

In addition, I do not detect any hints about a “fantastic theory”nor are any intended.

The paper is about the relationship between information theory, organisational practice and AI—it is about challenges in implementing AI/AGI successfully not just about AI/AGI in a research context. The information theory link is deliberately general (and applies much more widely than computer science)—many of the information theory ideas in the paper are only revealed with such a general view. As the paper says, AI and machine learning are forms of processing information and so the general ideas apply.

Reviewer 3 Report

The article develops a consistent argument about the informational limitations of Artificial Intelligence and machine learning. From the standpoint of Information Theory, it poses some challenges and theoretical difficulties in the way of Artificial General Intelligence developers. 

I think it is a good contribution for both the AI field and to this Special Issue, and it could be published in its present form.

Author Response

Thank you